# Current Management of Pleuropulmonary Blastoma: A Surgical Perspective

**DOI:** 10.3390/children6080086

**Published:** 2019-07-25

**Authors:** Samantha Knight, Tristan Knight, Amir Khan, Andrew J. Murphy

**Affiliations:** 1Southern Illinois University School of Medicine, Division of Surgery, Department of General Surgery, Springfield, IL 62702, USA; 2Division of Pediatric Hematology and Oncology, Department of Pediatrics, Children’s Hospital of Michigan, Detroit, MI 48201, USA; 3Department of Pediatrics, Wayne State University School of Medicine, Detroit, MI 48201, USA; 4Department of Surgery, St. Jude Children’s Research Hospital, Memphis, TN 38105, USA; 5Division of Pediatric Surgery, Department of Surgery, University of Tennessee Health Science Center, Memphis, TN 38105, USA

**Keywords:** pleuropulmonary blastoma, congenital pulmonary airway malformations, congenital cystic adenomatoid malformations, DICER1, multidisciplinary management, surgery

## Abstract

Pleuropulmonary blastomas (PPB) are pediatric, embryonal cancers of the lung parenchyma and pleural surfaces and are among the most common DICER1—related disorders. These tumors undergo evolution through several forms, allowing division into types I, Ir, II, and III, with correlates to the age of diagnosis and prognosis. We sought to provide a comprehensive review of the relevant literature describing the characteristics of these tumors and their multidisciplinary treatment, with an emphasis on surgical management. We describe the complementary roles of chemotherapy and surgery in the successful management of this disease. We discuss the timing of surgery and options for surgical approaches. We address the differentiation of PPB from congenital pulmonary airway malformation and the role of DICER1 testing for children with pulmonary cysts.

## 1. Introduction

Pleuropulmonary blastomas (PPB) are sarcomatous malignant pediatric tumors of the lung parenchyma and pleural surfaces which fall under the larger aegis of the DICER1-related disorders. Although PPB are the most common primary pediatric pulmonary malignancy, they are also extremely rare and occur at an incidence of 25–50 cases per year in the United States [1]. A PPB arises initially as a pulmonary cyst. Histologically, these cysts are lined with epithelium and also contain underlying sarcomatous mesenchyme. PPB undergo a well-documented progression through several forms which allows their division into types I, Ir, II, and III, with rough correlates to age at detection and prognosis [2]. A consensus has not been reached regarding the optimal surgical management of PPB. Therefore, we aim to provide a comprehensive review of the current literature describing the characteristics of PPB and multidisciplinary management.

## 2. Types of Pleuropulmonary Blastoma

Type I PPBs (purely cystic) appear as multiloculated, air-filled cysts with thin septa. The cysts are lined with benign respiratory epithelium and mesenchyme, with an underlying component of malignant mesenchymal cells that may have rhabdomyoblastic differentiation [2]. Approximately one third of PPBs are detected in this initial stage (type I), at a median age of 8 months, with 62% presenting in the first year of life, and 97% by 3 years of life. Type I PPB are most often unilateral, unifocal, peripheral, over 5 cm in size, and occur with a slight male predominance (57% of cases) [2].

Type I PPBs may then progress to become type II PPB (cystic and solid), defined by the appearance of a solid component derived from expansion of the malignant mesenchymal population of cells [1,2]. Type II PPBs account for approximately one-third of cases, with an equal male-to-female ratio, and present later than type I at a median age of 35 months (95% being diagnosed by 6.8 years of age). Type II PPB are very rarely seen prior to 12 months of age [2].

Type III PPBs (purely solid) present at a more advanced age, with a median age-of-diagnosis of 41 months, and do not appear to be seen before 12 months of age. Type III PPB are entirely comprised of tumor cells without intervening cystic spaces. The solid regions in types II and III PPBs are histologically similar, displaying a mixed, sarcomatous pattern. These sarcomatous cells may include interspersed foci of anaplasia, features of embryonal rhabdomyosarcoma, chondrosarcoma, or necrosis. Rarely, progression of necrosis may be sufficient to produce pseudocysts in some type III PPBs, but this does not ‘de-escalate’ a tumor to type II status [1,2,3]. Given this histologic similarity between types II and III, care must be taken to obtain adequate specimens during biopsy to capture the cystic component present in type II PPB (Figure 1).

Type I PPBs may alternatively progress to Type Ir (e.g., Type I regressed), which comprise 23% of the total type I population. Type Ir PPB have the same multilocular cystic appearance as type I, but without the interspersed primitive malignant cells. The ‘regressed’ moniker may be something of a misnomer because it is not truly known whether Type Ir PPBs ever possessed malignant components which then subsequently involuted. Type Ir PPB have a median age of diagnosis of 46.5 months compared to 8 months for Type I PPB. The small minority of patients who died of their disease following detection of a type I PPB (4.3% of type I and 1.4% of all PPBs), all occurred following progression to type II or type III, and no deaths were reported due to progression of a type Ir PPB [2]. A similar regression pattern has not been seen among confirmed cases of type II or III PPB.

## 3. Significance of DICER1 and Familial Tumor Predisposition Syndrome

The *DICER1* gene, located at chromosome 14q32.13, is a member of the ribonuclease III (RNase III) gene family, and its gene product (DICER1 protein) is a crucial component in the processing of microRNAs (miRNAs). MicroRNAs are short fragments of double stranded, non-coding RNA which post-transcriptionally regulate gene expression at the level of messenger RNA (mRNA) [4]. After transcription, and initial processing in the nucleus, miRNA enters the cytoplasm in a double-stranded ‘hairpin’ configuration, where it is recognized by DICER1 and cleaved into two single strands (mature miRNA) [5]. This mature, cleaved miRNA is then able to bind specific mRNA target sequences and induce mRNA degradation or inhibition of transcription. Inadequate mRNA suppression is known to be a potential driver of oncogenesis [6]. A homozygous absence of *DICER1* is not compatible with life in murine models [7]. Heterozygous loss of *DICER1* confers an increased risk of malignancy, but does not result in obvious phenotypic anomalies (Schultz et al., 2014) [5]. Germline loss of function mutations in *DICER1* have been identified in approximately 70%–80% of children with PPB. The presence of such a germline mutation defines DICER1 pleuropulmonary blastoma familial tumor predisposition syndrome (FTPS). Germline *DICER1* mutations are inherited in an autosomal dominant fashion, with 80% transmitted in this manner, and the remainder arising de novo [5]. The penetrance is rather low at approximately 10%–15%; therefore, many patients will never manifest any malignancy [8]. On the other hand, approximately 10% of the family members of a patient with PPB will display features related to the FTPS, and pulmonary ‘cysts’ which are in fact regressed PPBs (e.g., type Ir) are occasionally identified in adult family members [8,9]. It appears that this low penetrance may be due to the fact that a second ‘hit’ or mutation is required to impair the remaining normal copy of *DICER1* to sufficiently induce oncogenesis. Nearly all patients with PPB have an identifiable, second somatic mutation in *DICER1* in addition to their germline mutation. Children with PPB who do not carry a germline *DICER1* mutation most often display two somatic *DICER1* mutations [5]. In rare circumstances, PPB can occur without *DICER1* mutation [2]. Other genetic aberrations found in PPB without involvement of *DICER1* have included trisomies of chromosome 2 or 8, and variable deletions involving 17p—a region which includes *TP53* (17p13.1) [10,11]. Specific loss-of-function or inactivating mutations of *TP53* have also been identified, with some overlap between their occurrence and those of *DICER1* mutations [12].

The DICER1 FTPS increases the risk for multiple tumor types, and the presence of any such tumors, and certainly the presence of more than one such tumor, should trigger investigation for a germline *DICER1* mutation. The incidence of *DICER1* mutations in cystic nephroma, for instance, approaches 70%. However, almost all patients afflicted by both cystic nephroma and PPB demonstrate *DICER1* mutations [8]. Besides PPB, other associated tumors include ovarian sex cord-stromal tumors (including juvenile granulosa cell tumors, Sertoli-Leydig cell tumors, and gynandroblastomas), botyroid embryonal rhabdomyosarcoma, cystic nephroma, ciliary body medulloepithelioma, nasal chondromesenchymal hamartoma, pituitary blastoma, pineoblastoma, and thyroid neoplasias, including adenomas and thyroid cancer [5,13]. Less commonly seen malignancies in the FTPS include neuroblastoma, Wilms tumor, and medulloblastoma, although the presence of such malignancies need not necessarily engender a search for an underlying germline mutation [5,13]. An excellent review of the increased urogenital tumor-specific risks associated with germline *DICER1* mutations has recently been published, and reviews the incidence and associations in greater detail [8]; a more general review of all associated tumor types is also available [5].

## 4. Screening in Patients with Known *DICER1* Mutations

PPBs are sufficiently rare that screening of the general population is not warranted. Even among those with known *DICER1* mutations, only 4% of infants would be expected to have a detectable PPB [14]. Conversely, all patients with an identified PPB should be screened for *DICER1* mutations. Based on modelling data as proposed by the Committee on the Biological Effects of Ionizing Radiation, optimal surveillance consists of annual chest X-rays with suspicious findings further investigated by CT [14]. Given the autosomal dominant pattern of inheritance, screening for *DICER1* mutations is warranted in family members of patients with a germline mutation. The International PPB Registry (https://www.ppbregistry.org/) has recently released comprehensive recommendations for DICER1-related genetic testing, organ-specific surveillance, and prenatal management [13]. Specifically relating to PPB, *DICER1* testing is recommended for all individuals found to have a PPB of any type and for those with bilateral, septated, or multiple lung cysts in childhood. *DICER1* testing should be performed in all first-degree relatives of patients with *DICER1* mutations, with an emphasis on children under 7 years of age. However, in the approximately 20% of cases where the germline mutation has arisen de novo in the patient (parents test negative), testing of that patient’s siblings is not necessary as their risk is essentially that of the population at large (allowing for the slight possibility of germline mosaicism). While there is no role for prenatal genetic testing of the fetus, third trimester ultrasound is suggested in instances where either parent carries a known germline *DICER1* mutation. Following delivery and prior to 3 months of age, genetic testing for the pathogenic mutation should be performed. If the infant is found to have a *DICER1* mutation, a chest CT is recommended between 3 and 6 months of age, and an additional CT between 2–3 years of age (corresponding with the peak of type II and III PPB incidence). Otherwise, twice-yearly chest X-rays from birth until 7 years of age should be obtained, and thereafter, annually until 12 years of age. The value of continuing chest X-rays after 12 years of age is not known, and therefore, not recommended.

## 5. Relation between CPAM and Type I (Cystic) PPB

The congenital pulmonary airway malformations (CPAMs) are a group of non-malignant developmental anomalies which includes bronchogenic cysts, bronchopulmonary sequestration, congenital lobar emphysema, and congenital cystic adenomatoid malformations (CCAMs). Although sometimes used interchangeably, CCAMs refer to a specific subset or variety of CPAMs, and the two terms are not fully synonymous. Like the name of the disease entity itself, the classification of CPAMs has changed over time. Multiple systems exist, reflecting histopathological type, size, and appearance [15]. Formerly, controversy existed regarding whether a CPAM could degenerate into a PPB [16,17,18]. However, detailed genetic and immunohistochemical explorations have confirmed that distinct pathogenic mechanisms exist between these two disease entities [15,19]. While there are case reports of suspected malignant degeneration of CPAMs into sarcomas or other tumor types, in general CPAM and PPB are now regarded as separate entities [20]. Despite clarity regarding the separate pathogenesis of CPAM and PPB, the clinical differentiation of these two entities can be quite difficult. Type 4 CPAM can be confused with PPB and may appear similar on cross-sectional imaging. CPAMs are resected in symptomatic infants. However, for asymptomatic CPAMs, some advocate for observation rather than surgical resection [21,22].

A recent study developed a helpful clinical algorithm for the management of cystic pulmonary abnormalities in children with an emphasis on differentiating CPAM from PPB [23]. The clinical feature most strongly associated with CPAM compared to PPB was prenatal detection. Radiographic features strongly associated with CPAM included any area of pulmonary hyperinflation or presence of a systemic feeding vessel. Radiographic features associated with a high likelihood of PPB were multilobar or bilateral abnormalities, complex cysts, and mediastinal shift. *DICER1* screening was recommended for all patients with cystic pulmonary lesions who were originally intended to be observed [22].

Because PPB can transition from readily resectable cystic masses to potentially metastatic and lethal disease with solid components, we, in agreement with recommendations from the PPB Registry [13], would strongly advocate for *DICER1* germline testing to be performed in all pediatric patients with lung cysts. This is particularly important for cysts which are septated, multiple, bilateral, or identified in infancy. In addition, we would add the strong recommendation that those patients who are asymptomatic and being considered for a non-operative management strategy of CPAM undergo *DICER1* testing.

## 6. Complementary Roles of Surgery and Chemotherapy in the Treatment of PPB

Type I and Ir PPB have not been noted to metastasize and should be managed via complete resection with widely negative margins. The aggregate survival for type I and Ir is 91%, with all deaths following progression to more advanced tumor types. Adjuvant chemotherapy is not typically given for Type I PPB unless there are complicating circumstances such as intraoperative tumor spill, incomplete resection, or local invasion of adjacent structures. Sarcoma-based chemotherapy regimens including vincristine, dactinomycin, and cyclophosphamide (VAC) are most frequently utilized for PPB [2,24]. Incidental identification of purely cystic pulmonary lesions identified in adult patients with pathogenic *DICER1* mutations may also occur. In almost all instances, these are type Ir and, therefore, do not possess malignant potential [5,13]. If these older patients are asymptomatic and not otherwise experiencing morbidity related to their presumptive Type Ir PPB, removal is not explicitly indicated, and observation may be appropriate.

In type II and type III PPB, both systemic chemotherapy and surgical resection are critical components of treatment. The optimal chemotherapy regimen is not known, and there is no single ‘standard’ approach. Chemotherapy is typically based on sarcoma regimens—European centers often use a VAIAd approach (vincristine, dactinomycin, and ifosfamide alternating with vincristine, doxorubicin, and ifosfamide), and American centers most often use VAC or VACAd, sometimes appending these with platinum-containing agents and etoposide [1,2]. The first and thus-far only prospective clinical trial was opened in 2011 by the International Pleuropulmonary Blastoma Registry (IPPBR) and is currently ongoing. This study seeks to provide treatment guidelines for the various PPB types, thereby standardizing management by utilizing specific regimens with prospective review and efficacy analysis. In this trial, chemotherapy in type I PPB, if deemed necessary, is comprised of a 22-week regimen of four cycles of vincristine, actinomycin-D, and cyclophosphamide (VAC) and three subsequent cycles of vincristine and actinomycin-D (VA). Chemotherapy for type II and III PPB is comprised of a single arm of adjuvant/neoadjuvant ifosphamide, vincristine, actinomycin-d, and doxorubicin (IVADo), for a total of 36 weeks, (NCT01464606). Unlike type I/Ir PPB, type II and III may metastasize, most often to brain, but rarely to liver and bone. The histological appearance of metastatic lesions is relatively uniform, being comprised of sarcomatous (embryonal vs. undifferentiated) cells [2]. PET scan, bone scan, head CT, or brain MRI may, therefore, be appropriate tests in the metastatic workup of PPB.

Optimal timing for surgery and chemotherapy is dependent upon the type of PPB. Upfront resection is desirable in type I/Ir tumors, with the decision to append chemotherapy onto treatment, dependent upon such considerations as discussed above [5]. For type II/III tumors, consideration must be given to the anatomy of the tumor itself. If a tumor is amenable to complete early resection, this approach is preferred. Up-front complete PPB resection at diagnosis is desirable when at all possible, but pre-operative chemotherapy may be necessary in instances where this is not possible or presents an unacceptable risk to the patient. Although tumor size itself does not appear to correlate directly with outcome, Type II or III PPB larger than 10 cm may be exceedingly difficult to resect with negative margins. The European Cooperative Study Group for Paediatric Rare Tumors (EXPeRT) has identified a threshold of 10 cm as a reasonable cut-off point, above which up-front biopsy followed by neoadjuvant chemotherapy is a reasonable approach (Figure 2) [24]. Complete resection at diagnosis is associated with improved prognosis; however, no prospective studies examining the timing of resection and chemotherapy exist [24]. PPBs are relatively chemo-sensitive tumors, and a dramatic reduction in tumor volume between 40%–90% is often seen with neoadjuvant chemotherapy [1]. Close coordination between medical and surgical oncology services is advisable, however, the response to chemotherapy is often ephemeral and thus, rapid reoccurrence may be expected [1,5,24]. The current IPPBR protocol recommends resection occurs at Week 10, following three cycles of neoadjuvant chemotherapy and subsequent hematologic recovery.

## 7. Surgical Approaches to PPB

Type I PPB can be effectively cured by surgical resection with negative margins and no tumor spill. The typical approach for type I PPB is a generous pulmonary wedge resection. Central, hilar, or multifocal cystic lesions may require pulmonary lobectomy or more extensive resections. An open approach has been advocated to minimize the chance of tumor spill; however, smaller lesions may be approached thoracoscopically depending on the comfort level of the operating surgeon. Suspected type Ir lesions may be observed in teenagers and adults [13]. However, when resection is attempted, type Ir lesions should be approached in a similar fashion to other Type I PPB.

Surgical resection of type II and type III PPB may require anything from a wedge resection to lobectomy or pneumonectomy to achieve negative margins if possible. These operations should be performed in an open fashion. When performing a lobectomy for type II or III PPB, involved pleural surfaces should be resected en bloc with the primary tumor and involved pulmonary lobe. For large type II or III PPB with extensive pleural spread, an extrapleural pneumonectomy may be required to achieve local control (Figure 2) [25,26,27]. Extrapleural pneumonectomy entails resection of the pleural surfaces, pericardium, phrenic nerve, and diaphragm with intrapericardial control and division of the pulmonary hilar vessels. The pericardial defect is reconstructed using a fenestrated patch to prevent cardiac herniation into the pleural space and to prevent accumulation of pericardial fluid and tamponade physiology. The diaphragmatic defect is reconstructed using an unfenestrated patch. It should be recognized that pneumonectomy is associated with a high rate of post-operative complications and post-pneumonectomy syndrome in the pediatric age group due to the mobility of mediastinal structures in children. Post-pneumonectomy syndrome is caused by mediastinal shift with critical compression or kinking of the residual mainstem bronchus, trachea, or great vessels. Tissue expanders can be used to mitigate complications related to post-pneumonectomy syndrome, although their use in the pediatric age group is limited [28]. Neoadjuvant chemotherapy should be utilized in cases of large type II or III PPB to minimize the radical surgical approach needed to achieve local control. There has been no demonstrated benefit of radiotherapy in this disease. In rare cases, PPB have demonstrated endobronchial or pulmonary vascular extension [29]. Therefore, echocardiography and bronchoscopy should be utilized in cases of advanced PPB to increase awareness of the disease anatomy and plan the appropriate surgical approach.

## 8. Summary

*DICER1* testing is recommended for all individuals with PPB. *DICER1* testing should be strongly considered for those with bilateral, septated, or multiple lung cysts in childhood, particularly if a non-operative management strategy is being employed. Optimal timing for surgery is dependent on the type of PPB. Upfront resection with widely negative margins is desirable and typically achievable for type I PPB. Chemotherapy is typically not indicated for Type I PPB unless there is intraoperative tumor spill or positive margins. Surgery and chemotherapy have complementary roles for Type II and III PPB. For type II/III tumors, biopsy followed by neoadjuvant chemotherapy can be utilized for large tumors predicted to be difficult to resect with negative margins (a reasonable cutoff is >10 cm). PPB are relatively chemosensitive tumors, and a dramatic reduction in tumor size may be seen with neoadjuvant chemotherapy. Chemotherapy based on sarcoma regimens is administered to all patients with Type II or III PPB.

## Figures and Tables

**Figure 1 children-06-00086-f001:**
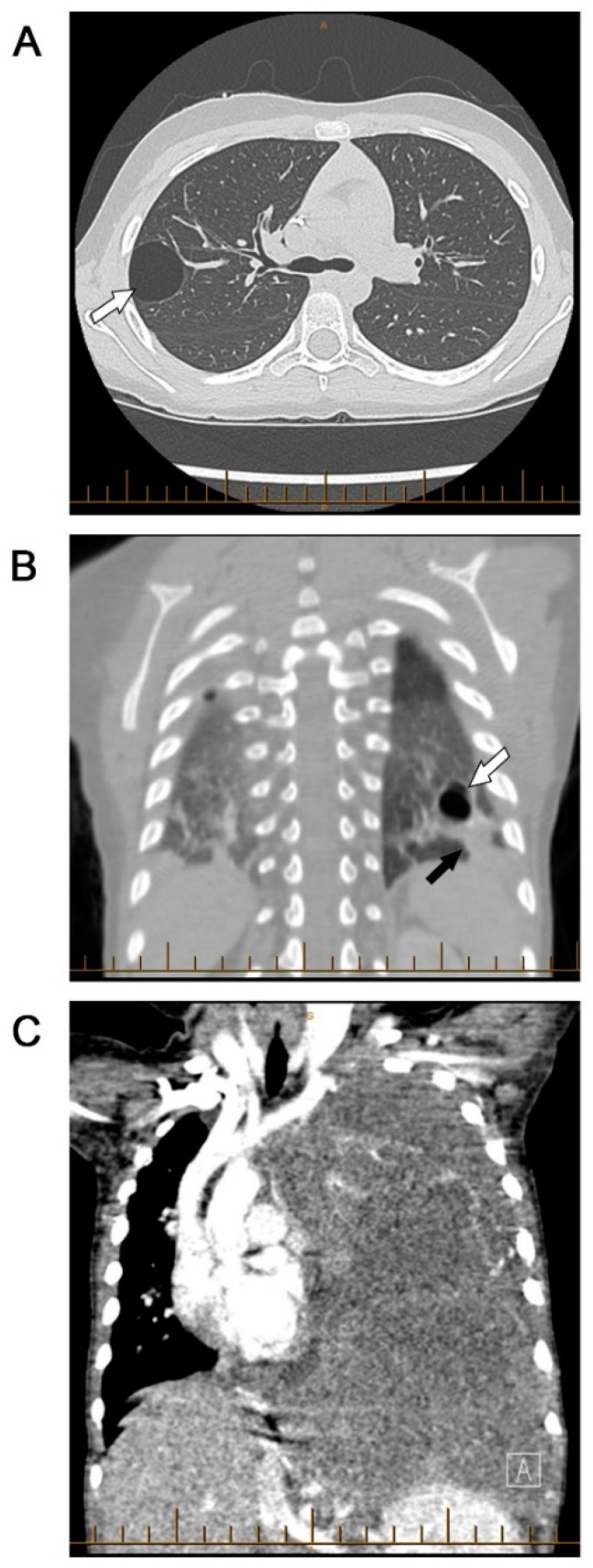
Types of pleuropulmonary blastomas (PPB) (**A**) Type I purely cystic PPB (white arrow). (**B**) Type II PPB with cystic (white arrow) and solid (black arrow) components. (**C**) Type III purely solid PPB occupying the entire left hemithorax.

**Figure 2 children-06-00086-f002:**
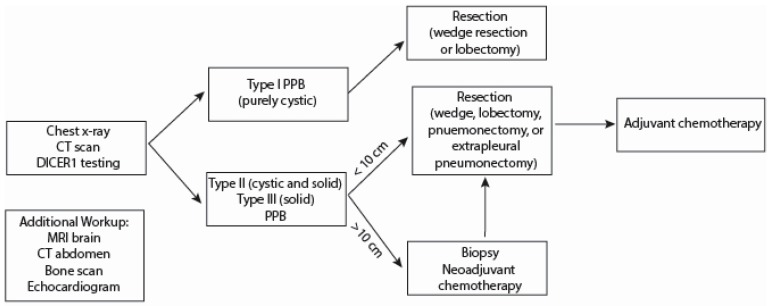
Proposed management algorithm based on type and complexity of PPB. Note: Type I PPB may also be >10 cm in size, but neoadjuvant chemotherapy is not recommended for type I PPB because the purely cystic morphology is not predicted to respond in a manner that would make obtaining negative surgical margins any easier. Furthermore, chemotherapy can be spared entirely if Type I PPB are resected with negative margins, regardless of lesion size.

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
