# Peer review of "Current Management of Pleuropulmonary Blastoma: A Surgical Perspective"

_children, 2019, doi:10.3390/children6080086_

Reviewer 1 Report

The authors present a brief review of the PPB literature with information from several relevant sources including chapters and articles. Unfortunately however, there are several places with misleading or out of date information including the section on CPAM vs. PPB. In addition, although this is described as a surgical review, one of the most definitive papers about the role of surgical management is not included. Little evidence is provided to support the specific surgical recommendations. Specific statements such as "tumors larger than 10cm in size are uniformly type II or III" are not correct.

Author Response

Reviewer 1: The authors present a brief review of the PPB literature with information from several relevant sources including chapters and articles. Unfortunately however, there are several places with misleading or out of date information including the section on CPAM vs. PPB.

 Response: The CPAM versus PPB section has been updated to emphasize that these disease entities are pathologically distinct and do not share the same genetic basis. However, we have emphasized that their differential diagnosis and appropriate clinical management can be difficult and we have emphasized current literature to address this scenario. We have specifically focused on the radiographic and clinical features which differentiate the two disease entities and the role of DICER1 germline testing for children with pulmonary cysts.

 Reviewer 1: In addition, although this is described as a surgical review, one of the most definitive papers about the role of surgical management is not included.

Response: Because the reviewer did not indicate any specific information whatsoever regarding this definitive paper, we were unable to include a reference to this work in the revised version of the manuscript. We would be happy to do so if any information can be provided.  

 Reviewer 1: Little evidence is provided to support the specific surgical recommendations. Specific statements such as "tumors larger than 250px in size are uniformly type II or III" are not correct.

 Response: This statement has been removed from the manuscript. The fact that Type I PPB can be larger than 10 cm has been included in the legend to Figure 2. The 10 cm “cutoff” for biopsy and neoadjuvant chemotherapy has been more clearly depicted as a “reasonable” guideline to assist in surgical decision making rather than a strict, abrupt cutoff.  We believe the recommendations overall have been appropriately referenced.

Reviewer 2 Report

Outstanding review of PPB that covers the genetic predisposition and therapy options. Highly informative article. 

I find this work succinct and informative for what we know about this rare cancer and its protean forms.

Author Response

Reviewer 2: I find this work succinct and informative for what we know about this rare cancer and its protean forms.

 Response: Thank you for your encouraging review of our manuscript.

Reviewer 3 Report

The authors present a summary of PPB focusing on the surgical aspect. The data is here but think there are a few areas for improvement.

the sentence structure throughout the entire manuscript is complex and difficult to follow, Suggest decreasing the commas by half and instead break the sentences down into simpler and more direct statements. Currently almost need a roadmap for the sentence structure. This is not a complex topic so don't make the reader work hard to figure out what you are trying to say.

try not to make it sound like it is surgery vs chemotherapy in the manuscript. They are complimentary and each should be employed when appropriate. 

Try not to repeat sentences verbatim between sections in the paper.

In section 2 please break down the different types into different paragraphs

rather than discussing a "less evolved" would suggest just using non-malignant less evolved sounds similar to undifferentiated

Your discussion of Ir is confusing about what is the etiology when you just state "the former explanation". Go into more detail describing what pathogenesis you believe. Again in some ways this gets back to the complex sentence structure.

Same comment about unclear complexity regarding DICER evaluation and the relationship between DICER and tumor.

line 179 never heard of "contradictory-such". In addition line 269 seems incomplete.

Need a summary section at the end of the paper.

Author Response

Reviewer 3: The authors present a summary of PPB focusing on the surgical aspect. The data is here but think there are a few areas for improvement.

the sentence structure throughout the entire manuscript is complex and difficult to follow, Suggest decreasing the commas by half and instead break the sentences down into simpler and more direct statements. Currently almost need a roadmap for the sentence structure. This is not a complex topic so don't make the reader work hard to figure out what you are trying to say.

 Response: We acknowledge that the sentence structure in the original draft was exceedingly complex and detracted from the straightforward message of the content. We have thoroughly revised the manuscript with this reviewer’s advice in mind. Specifically, we have made a concerted effort to decrease the number of commas leading to complex sentence structure throughout the manuscript. Sentences have been broken up into direct remarks when possible.  

 Reviewer 3: try not to make it sound like it is surgery vs chemotherapy in the manuscript. They are complimentary and each should be employed when appropriate. 

 Response: We have revised the manuscript to emphasize the complementary roles of surgery and chemotherapy in the management of pleuropulmonary blastoma. We have renamed section 6 “Complementary Roles of Surgery and Chemotherapy in the Treatment of PPB” to emphasize this point. We have eliminated any language that may have seemed like surgery and chemotherapy were pitted against one another in the manuscript.

 Reviewer 3: Try not to repeat sentences verbatim between sections in the paper.

 Response: The abstract has been rewritten to avoid repeated sentences in the manuscript. The summary section uses different language from prior sentences addressing these points.

 Reviewer 3: In section 2 please break down the different types into different paragraphs

 Response: Each PPB type (I, II, III, Ir) is now described in a separate paragraph in section 2 in the revised version of this manuscript.

 Reviewer 3: rather than discussing a "less evolved" would suggest just using non-malignant less evolved sounds similar to undifferentiated

 Response: We have eliminated the term “evolved” from the manuscript.

 Reviewer 3: Your discussion of Ir is confusing about what is the etiology when you just state "the former explanation". Go into more detail describing what pathogenesis you believe. Again in some ways this gets back to the complex sentence structure. Same comment about unclear complexity regarding DICER evaluation and the relationship between DICER and tumor.

 Response: We have clarified what is known about the pathogenesis about Type Ir pleuropulmonary blastoma and the recommendations for DICER1 evaluation. The confusing nature of the initial version was likely due to excessively complex sentence structure. We believe these sections are clearer now that we have revised the sentences with the reviewer’s recommendations in mind.

 Reviewer 3: line 179 never heard of "contradictory-such". In addition line 269 seems incomplete.

 Response: This term has been eliminated from the revised version. We have fixed the sentence from line 269 of the original submission.

 Reviewer 3: Need a summary section at the end of the paper.

 Response: We have added a summary section at the end of the paper.